# Changes of Conformation in Albumin with Temperature by Molecular Dynamics Simulations

**DOI:** 10.3390/e22040405

**Published:** 2020-04-01

**Authors:** Piotr Weber, Piotr Bełdowski, Krzysztof Domino, Damian Ledziński, Adam Gadomski

**Affiliations:** 1Atomic and Optical Physics Division, Department of Atomic, Molecular and Optical Physics, Faculty of Applied Physics and Mathematics, Gdańsk University of Technology, ul. G. Narutowicza 11/12, 80-233 Gdańsk, Poland; piotr.weber@pg.edu.pl; 2Institute of Mathematics and Physics, UTP University of Science and Technology, Kaliskiego 7, 85-796 Bydgoszcz, Poland; agad@utp.edu.pl; 3Institute of Theoretical and Applied Informatics, Polish Academy of Sciences, Bałtycka 5, 44-100 Gliwice, Poland; kdomino@iitis.pl; 4Faculty of Telecommunications, Computer Science and Technology, UTP University of Science and Technology, 85-796 Bydgoszcz, Poland; damian.ledzinski@utp.edu.pl

**Keywords:** Flory–De Gennes exponent, conformation of protein, albumin, non-gaussian chain, non-isothermal characteristics, Fisher’s test, Kullback–Leibler divergence

## Abstract

This work presents the analysis of the conformation of albumin in the temperature range of 300K–312K, i.e., in the physiological range. Using molecular dynamics simulations, we calculate values of the backbone and dihedral angles for this molecule. We analyze the global dynamic properties of albumin treated as a chain. In this range of temperature, we study parameters of the molecule and the conformational entropy derived from two angles that reflect global dynamics in the conformational space. A thorough rationalization, based on the scaling theory, for the subdiffusion Flory–De Gennes type exponent of 0.4 unfolds in conjunction with picking up the most appreciable fluctuations of the corresponding statistical-test parameter. These fluctuations coincide adequately with entropy fluctuations, namely the oscillations out of thermodynamic equilibrium. Using Fisher’s test, we investigate the conformational entropy over time and suggest its oscillatory properties in the corresponding time domain. Using the Kruscal–Wallis test, we also analyze differences between the root mean square displacement of a molecule at various temperatures. Here we show that its values in the range of 306K–309K are different than in another temperature. Using the Kullback–Leibler theory, we investigate differences between the distribution of the root mean square displacement for each temperature and time window.

## 1. Introduction

The dynamics of proteins in solution can be modeled by a complex physical system [1], yielding many interesting effects such as the competition and cooperation between elasticity and phenomena caused by swelling. Biopolymers, such as proteins, consist of monomers (amino acids) that are connected linearly. Thus, a chain seems to be the most natural model of such polymers. However, depending on the properties of the polymer molecule, modeling it as a chain may give more or less accurate outcomes [2]. The most straightforward approach to the polymer is the freely jointed chain, where each monomer moves independently. The excluded volume chain effect, which prevents a polymer’s segments from overlapping, is one of the reasons why the real polymer chain differs from an ideal chain [3]. The diffusive dynamics of biopolymers are discussed further in [4] and the references therein, and the subdiffusive dynamics are discussed in [5,6]. For a given combination of polymer and solvent, the excluded volume varies with the temperature and can be zero at a specific temperature, named the Flory temperature [7]. In other words, at the Flory temperature, the polymer chain becomes nearly ideal [8]. However, this parameter is global, and a more detailed description of the dynamics is required to perform additional detailed analyses. Our work concentrates on an analysis of albumin (shown in Figure 1) [9]. Albumin plays several crucial roles as the main blood plasma protein (60% of all proteins): controlling pH, inducing oncotic pressure, and transporting fluid. This protein (of nonlinear viscoelastic, with characteristics of rheopexy [10]) plays an essential role in the process of articular cartilage lubrication [11] by synovial fluid (SF). SF is a complex fluid of crucial relevance for lowering the coefficients of friction in articulating joints, thus facilitating their lubrication [12]. From a biological point of view, this fluid is a mixture of water, proteins, lipids, and other biopolymers [13]. As an active component of lubricating mechanisms, albumin is affected by friction-induced increases of temperature in synovial fluid. The temperature inside the articular cartilage is in the range of 306K–312K [14], and changes occurring inside the fluid can affect the outcome of albumin’s impact on lubrication [10]. Albumin can help to regulate temperature during fever. Therefore, a similar mechanism could take place during joint friction, increasing SF lubricating properties as an efficient heat removal. We analyze this range of temperature: from 300K–312K.

Let us move to the technical introduction of albumin dynamics. Consider a chain that was formed by N+1 monomers, assuming the following positions, designated by $\left\{{\vec{r}}_{0},{\vec{r}}_{1},\ldots ,{\vec{r}}_{N}\right\}$. We denote by ${\vec{\tau }}_{j}={\vec{r}}_{j}-{\vec{r}}_{j-1}$ a vector of a distance between two neighboring monomers. While considering the albumin, we have N+1=579 amino acids. For the length of the distance vector ${\vec{\tau }}_{j}$, we take the distance between two α-carbons in two consecutive amino acids, hence we have $|{\vec{\tau }}_{j}|=\tau =3.8$ [Å] [15]. Using such a notation, we can define the end-to-end vector $\vec{R}$ by the following formula:(1)$$\vec{R}={\vec{r}}_{N}-{\vec{r}}_{0}=\sum_{j=1}^{N}{\vec{\tau }}_{j}.$$In the case of an ideal chain, the distance vectors (for all *N*s) are completely independent of each other. This property is expressed by a lack of correlation between any two different bonds, that is [16]:(2)$$\langle {\vec{\tau }}_{i}\cdot {\vec{\tau }}_{j}\rangle =\tau^{2}\delta_{ij},$$
where $\delta_{ij}$ is the Kronecker delta. Equations (Equation 1) and (Equation 2) imply that a dot self-product of vector $\vec{R}$ reads:(3)$$\langle \vec{R}\cdot \vec{R}\rangle =\langle R^{2}\rangle =\sum_{i=1}^{N}\sum_{j=1}^{N}\langle {\vec{\tau }}_{i}\cdot {\vec{\tau }}_{j}\rangle =N\tau^{2},$$
where *R* is the length of the vector $\vec{R}$. The value of *R* is measured by calculating the root mean square end-to-end distance of a polymer. The *R*’th value is assumed to depend on the polymer’s length *N* in the following manner [17]:(4)$$R∼\tau N^{\mu },$$
where μ is called the size exponent. If a chain is freely joined, it appears that μ=1/2. Relation (Equation 4) suggests we consider the dynamics of the albumin protein in some range of temperature and look for dependence between the temperature and exponent μ. Applying the natural-logarithm operation to (Equation 4) yields:(5)$$\mu ≃\frac{\log R-\log\tau }{\log N}.$$In Equation (Equation 4) we use a ∼ symbol to emphasize that we use $\langle R^{2}\rangle$ to obtain *R*. In the case discussed here, both parameters τ and *N* (the number of amino acids) are constant. Importantly, using molecular dynamics simulations, one can obtain the root mean square end-to-end distance for various temperatures. The idea of the chain points to its description by backbone angles and dihedral angles. Backbone angles are angles between three α-carbons occurring after each other and signed by ϕ. The second angle, designated by ψ, is a dihedral angle [18]. Using this parametrization, we can obtain an experimental distribution of probabilities. Such distributions have features that may display some time-dependent changes.

These time-dependent changes may have an utterly random course, but sometimes periodic behavior can be seen. This property can be analyzed using a particular statistical test [19]. There is an assumption that the time series sampling is even, and the test treats it as a realization of the stochastic model evolution, which consists of specific terms of the form:(6)$$y_{n}=\beta \cos\left(n\omega +\alpha \right)+\epsilon_{n},$$
where β≥0, n=0,1,2,…,N, 0<ω<π, −π<α≤π is uniformly distributed in (−π;π] phase factor and $\epsilon_{n}$ is an independent, identically distributed random noise sequence. The null hypothesis is that amplitude β is equal zero: ($H_{0}$: β=0) against the alternative hypothesis that $H_{1}$: β≠0.

A periodogram of a time series ($y_{0},y_{1},\ldots ,y_{N}$) is used to test the above mentioned hypothesis and it is defined by the following formula:(7)$$I\left(\omega \right)=\frac{1}{N}|\sum_{j=0}^{N}\exp\left(-i\omega j\right)y_{j}{|}^{2}.$$
where in general, ω∈[0,π] and i is the imaginary unit. For a discrete time series, a discrete value of ω from [0,π] is being used:(8)$$\omega_{k}=\frac{2\pi k}{N},\;k=0,1,\ldots ,N,$$
which are known as Fourier frequencies.

The periodogram (for a single time series) can contain a peak. The statistical test should give us information about the significance of this selected peak. It is measured by the *p*-value parameter. The statistic which is used in this test has the form:(9)$$g=\frac{max_{1\leq i\leq q}I\left(\omega_{i}\right)}{\sum_{i=1}^{q}I\left(\omega_{i}\right)},$$
where: $I(\omega_{i})$ is the spectral estimate (the periodogram) evaluated at Fourier frequencies and $q=\left[(N-1)/2\right]$ is the entire of the number (N−1)/2. The *p*-value can be calculated according to the formula [20]:(10)$$P\left(g\geq x\right)=\sum_{j=1}^{p}{\left(-1\right)}^{j-1}\left({}_{j}^{n}\right){\left(1-jx\right)}^{n-1}$$
where *p* is the largest integer less than 1/x. In our analysis, we use this framework to evaluate a *p*-value for a given time series of entropy. If we get $g^{*}$ from the calculation of *g*, then Equation (Equation 10) gives a *p*-value for probability that the statistic *g* is larger than $g^{*}$.

Proteins can be characterized by various parameters that can be obtained from simulations, such as the earlier mentioned 〈R〉 or μ. Another meaningful parameter is the RMSD (root mean squared displacement), which can be calculated in every moment of the simulation and is defined as:(11)$$RMSD=\sqrt{\frac{\sum_{i=1}^{n}R_{i}\cdot R_{i}}{n}}$$
where: $R_{i}$ is a position of the *i*-th atom. In this way, one can obtain a series of this parameter as a function of time for the further processing. The drawing procedure is realized at the beginning of the simulation from the area of initial values. The initial condition for the albumin molecule drawing is simulated for a given temperature.

We can then group molecules according to mentioned parameters (〈R〉, μ, RMSD) and obtain five sets for 300K, 303K, 306K, 309K, and 312K, and compare these parameters’ values. We decided to use the Kruscal–Wallis statistical test because it does not assume a normal distribution of the data. This nonparametric test indicates the statistical significance of differences in the median between sets. A statistic of this test has the form [21]:(12)$$H=\frac{12}{N(N+1)}\sum_{i}^{k}\frac{rs_{i}^{2}}{n_{i}}-3\left(N+1\right),$$
where $rs_{i}$ is a sum of all rank of the given sample, *N* represents the number of all data in all samples, and *k* is the number of the groups (set into temperature). We compare the *H* value of this statistic to the critical value of the $\chi^{2}$ distribution. If the statistic crosses this critical value, we use a nonparametric multi-comparison test—the Conover–Iman. The formula which we employ has the form [22]: (13)$$|\frac{rs_{i}}{n_{i}}-\frac{rs_{j}}{n_{j}}|>t_{1-\alpha /2}\sqrt{\frac{\left(S_{r}-C\right)\left(N-1-H\right)}{n_{i}n_{j}\left(N-k\right)\left(N-1\right)}}$$
where $S_{r}$ is the sum of all square of ranks, $t_{1-\alpha /2}$ is a quartile from the Student *t* distribution on N−k degrees of freedom, and *C* has the value:(14)$$C=0.25N{\left(N+1\right)}^{2}.$$The multi-comparison test gives the possibility of clustering sets of molecules into those between which there is a statistically significant difference and those between which there is no such difference.

Statistical tests describe differences between medians of the considered sets of molecules. However, they are not informative about differences between distributions of probabilities. In our work, we use the Kullback–Leibler divergence, which is used in information theory, to compare two distributions. The Kullback–Leibler divergence between two distributions *p* and the reference distribution *q* is defined as:(15)$$KL\left(p,q\right)=\sum_{i}p_{i}\log\left(\frac{p_{i}}{q_{i}}\right).$$

The Kullback–Leibler divergence is also called the relative entropy of *p* with respect to *q*. Divergence defined by (Equation 15) is not symmetric measure. Therefore, sometimes one can use a symmetric form and obtain the symmetric distance measure [23]:(16)$$KL_{dist}\left(p,q\right)=\frac{KL(p,q)+KL(q,p)}{2}.$$

The measure defined by (Equation 15) is called relative entropy. It is widely used, especially for classification purposes. In our work, we also use this measure to recognize how a given probability distribution is different from a given reference probability distribution. We can look at this method as a complement to the Kruscall–Wallis test. The Kruscall–Wallis test gives information about the differences between medians, while the Kullback–Leibler divergence gives us information about the differences in distributions.

## 2. Results

We begin the analysis by examining the global dynamics of molecules at different temperatures using the Flory theory, briefly described in the introduction. To estimate the exponent μ from definition in Formula (Equation 5), we collect outcomes from a molecular dynamics simulation in the temperature range of 300K–312K. We use a temperature step of 3K. Such a range of temperature was chosen because we operated within the physiological range of temperatures. During the simulation, we can follow the structural properties of molecules in the considered range of temperatures. In the albumin structure, we can distinguish 22 α-helices, which are presented in Figure 1. Molecules in the given range of parameters display roughly similar structures. Therefore, we chose to study their dynamics in several temperatures. One important property, which we can follow during simulations, is a polymer’s end-to-end distance vector as a function of time. We perform the analysis in two time windows, first, between 0–30 ns and, second, between 70–100 ns. For each time window, we obtained the root mean square end-to-end distance of this quantity, whose values are presented in Table 1:

We can see that the root mean square length of the end-to-end vector for albumin varies with temperature. To obtain statistics, we can perform the Kruscal–Wallis test of this variation. In our work, values of the statistics in the test are treated as continuous parameters. We do not specify a statistical significance threshold. In Table 2, in the second column, we can see the values of statistics calculated according to Formula (Equation 12). The *H* parameters from the Kruscal–Wallis test have small values for this test, so the results tend to support the hypothesis about equal medians.

The Kruscal–Wallis test only gives information about differences between medians of the sets, where each set represents one temperature and elements of the set are calculated as root mean square end-to-end vectors over time. In the analysis, for one temperature, we had the set of root mean square end-to-end vectors, and in the next step, we calculated a numerical normalized histogram. Each element of this set corresponds to the single realization of the experiment. The values of the Kullback–Leibler divergence for mean end-to-end signals in the time window 0–30 ns are presented in Table 3. In the first column is the reference distribution for the temperature range. We can see that the biggest value is for the distribution of the temperature 312K in the reference distribution 309K.

We can calculate the Kullback–Leibler distance according to Formula (Equation 16). The results are presented in Table 4. This measure indicates that, between distributions, the biggest differences are between the distribution for 309K and the distribution for 312K.

We can perform the same analyses for the window 70–100 ns. The obtained values of the Kullback–Leibler divergence, which are presented in Table 5. One can see that the biggest difference is between the distribution for 312K and the distribution for 303K. One can see that the maximum has changed and its value is bigger than the maximum for the 0–30 ns window.

For the symmetric case, results are presented in Table 6. We can see that a symmetrical form of the measure is also between the distribution for 312K and the distribution for 303K.

Table 7, Table 8, Table 9 and Table 10 give the different statistical approaches, without the averaging over the time step). We present there all end-to-end signals for one temperature, and for each moment of time, we calculate the average only over all simulated molecules. In the next step, we calculate normalized histograms. After obtaining numerical distributions, we calculate the Kullback–Leibler divergence between distributions of mean end-to-end signals. The results for the time window 0–30 ns are presented in Table 7.

We can see that the large value of the Kullback–Leibler divergence appears for the distribution for temperature 306K, where the distribution for temperature 300K is a reference distribution. Calculations for the Kullback–Leibler distance for mean end-to-end signals in the time window 0–30 ns are presented in Table 8. The results coincide with the measure of Kullback–Leibler divergence.

Following the copula approach presented in [24], we performed the bivariate histograms of some signals to analyse the type of cross-correlation between signals. For the selected example of the mean end-to-end signals in the 0–30 ns time window see Figure 2, while for the 70–100 ns time window, see Figure 3. Observe that for the 0–30 ns case, we have simultaneous “high” events, which refer to the “upper tail dependency”. After the simulation time has passed, this dependency is diminished. The copula with the “upper tail dependency”, such as the Gumbel one, can provide the proper model of the mean end-to-end signal for the initial simulation time window. For the later simulation time window, we should look for the copula with no “tail dependencies”, such as the Frank or Gaussian one.

A similar calculation has been performed for the next time window 70–100 ns. In Table 9, we collect results for Kullback–Leibler divergence for mean end-to-end signals in the time window 70–100 ns.

Its maximal value is between the distribution for 306K and the distribution for 303K as a reference distribution.

In Table 10 we can see that the biggest value of the Kullback–Leibler distance appears between the distributions for 306K and 309K. Comparing this result with the previous one, we see that the maximal Kullback–Leibler distance for the window 0–30 ns changes its place.

In Table 7, Table 8, Table 9 and Table 10 the diagonal terms are equal 0, since we are comparing a distribution of end-to-end distributions of raw data for the same time window and temperature. In Table 11 we present selected Kullback–Leibler distances for distributions of raw data end-to-end signals in the time window 70–100 ns and 306K temperature.

As two different statistical approaches can not unambiguously distinguish between temperatures (or their subsets), we can conclude that dynamics of the system at this temperatures range is roughly similar. This will be approved by the analysis of the the Flory–De Gennes exponent.

Using Equation (Equation 5), we estimate values of the Flory–De Gennes exponent for the given protein chain. Results are presented in Table 12. We can see that the values of the Flory–De Gennes exponents do not differ too much within the examined range of temperature, and they all seem to attain a value of ca. 0.4. The power-law exponents for oligomers span a narrow range of 0.38–0.41, which is close to the value of 0.40 obtained for monomers [25,26,27].

Detailed values for two time windows are presented in Table 13. We can see that from a global point of view and Flory theory, the situation is quite uniform for all initial conditions and is within the considered range of temperature. Thus, in our modeling, we see an effect of elasticity vs. swelling. This competition is preserved for the examined temperature range.

The *H* parameters from the Kruscal–Wallis test have small values for this test, so the results tend to support the hypothesis about equal medians.

The distribution of backbone dihedral angles carries information of the molecule’s dynamics, as they are tightly connected to its elasticity. Using simulations, we can determine the angle ϕ and the angle ψ (see Introduction). Next, we can determine energies that depend on these angles by employing Formula (Equation 17). In Figure 4 and Figure 5, we present the logarithm of the angle energy component for two temperatures. Figure 4 presents values in a period of time 0–10 ns, and Figure 5 presents values in a period of time 90–100 ns. We can see that these values fluctuate around the mean value. Such dynamics are similar for all temperatures, but the mean energy rises slightly with increasing temperature. In Figure 5, the blue line presents a logarithm of the mean value of angle energy and the gray line presents values of the standard deviation. We can see that changes in energy are much more evident.

Different outcomes come from an analysis of the energy component connected with the dihedral angles ψ. See Figure 6 (blue lines) in double logarithmic scale, and note the linear regression lines. We present a period of time from 1 ns to 10 ns.

Simulations also show that for further times in the log-log scale, these values exhibit almost no change with time.

In Figure 7, we present regression parameters from Figure 6.

The dynamics of proteins are governed mainly by non-covalent interactions [28]. Therefore, it would be useful to study other components of energy, such as the Coulomb component and the van der Waals component (see Equation (Equation 17) and the discussion following it). The Coulomb component oscillates around the almost constant trend present for every temperature. However, the mean value of these oscillations increases with the temperature. The opposite situation appears for the van der Waals component. Similar to the Coulomb component, for each selected temperature, it follows an almost constant trend. However, when the temperature increases, the mean value of such oscillations decreases. The Coulomb component and the van der Waals component obey simple dynamics. Hence, we can move to a further discussion of more interesting dihedral and angular parts of the energy.

Simulations of albumin dynamics can be used to obtain frequency distributions of angles ϕ and ψ. An exemplary 2 D histogram of such a distribution is presented in Figure 8. We can see that most data are concentrated in a small area of space of angles. Similar behavior can be observed for other temperatures and simulation times. This is because the structure of the investigated protein is quite rigid, and there appears to be only a little angular movement. Nevertheless, we can observe variations of the conformational entropy, estimated with such a histogram (see Figure 9).

According to formulas connecting the probability distribution of angles and entropy [17], we can follow changes of entropy in time and temperature. Values of conformational entropy for various temperatures and with simulation time 0–30 ns are presented in Figure 9, where the blue line represents the course of the mean entropy value over time. The calculations take into account nine simulations, and one dot presents one arithmetic mean of entropy. Here, we can see that during simulations, some oscillations can appear. Therefore, for each temperature, we perform Fisher’s test. Results of this test are presented in Table 14.

A simple comparison of the *p*-values in Table 14 shows that the lowest value is located around 306K degrees. According to Formula (Equation 6), mean entropy is supposed to be treated as stochastic, i.e., partially deterministic and partially random. In our work, we treat the *p*-value as a parameter, which gives us information about the tendency on maintaining the truth of the hypothesis: β=0. If the *p*-value is lower than in another case, then it presents a stronger tendency in favor of the alternative hypothesis: β≠0.

Much more detailed calculations of the *p*-values for this case are presented in Table 15. We can see that at 306K, there is a local minimum of the mean *p*-value. It indicates that at 306K, the tendency towards the alternative hypothesis: β≠0 is stronger than it is at 303K and 309K. However, for 300K, the mean *p*-value parameter is the smallest. The mean *p*-value parameter is almost the same for 306K and 312K. This means that both temperatures exhibit nearly the same tendency for the hypothesis: β≠0. Such regularities can be seen in Figure 9.

The same analyzes can be done on the data presented in the image.

The same analyzes can be done on the data presented in Figure 10. The results of these analyzes, in relation to individual time series of entropy, are presented in the Table 16. For these data, we also observe the local minimum of the *p*-value parameter for 306K. We have a maximum at 303K. Thus, the tendency towards the hypothesis β=0 is strongest here in comparison to the rest of the analyzed temperatures. This maximum value of the *p*-value parameter for 303K is also present for the previous window.

Formula (Equation 11) can be calculated for each moment of simulation, so we obtain a time series. To get one parameter for one molecule, we calculate a mean value of this parameter for each molecule. In the next step, we can perform the same statistical considerations to derive values of the *H* statistic. For the time window 0–30 ns, the value of the Kruscal–Wallis test gives H=9.41; however, for the time window 70–100 ns, the value of this statistic is H=12.52. Because we treat the statistic as an ordinary parameter, we do not specify whether results are significantly different. In our approach, we conclude that the tendency for the alternative hypothesis about not equal medians is much higher for the time window 70–100 ns. Since this tendency is more significant in the second case than in the previous one, we decided to perform a multi-comparison test. We chose the Conover–Iman multi-comparison test. We treated $t_{\alpha -1/2}$ as a parameter, which depends on an assumed statistical significance. Therefore in Table 17, we only give the left side of Equation (Equation 13).

Here, we have a comparison between medians for different sets. Using values from Table 17 and calculating the right sight of Equation (Equation 13) for p=0.05, we can state that there are sets that differ from another cluster of sets. The cluster of sets for temperatures 306K, 309K, and 312K differ from the set for temperature 303K. Unfortunately, the set for temperature 300K is between these two.

We can compare the above statistical results to the other methods such as the Kullback–Leibler distance, which is presented in the Introduction. In Table 18, we have values of this quantity. We can see that the biggest value is between the distribution for 300K and the distribution for 306K as a reference distribution.

The calculation of the Kullback–Leibler distance is presented in Table 19. We can conclude that the 306K case differs most from all other cases, so the dynamics of the system in the temperature range of 306*K*–309*K* appears to differ most from the dynamics of the system in other temperature ranges.

The calculations of Kullback–Leibler divergence and distance for the time window 70–100 ns are presented in Table 20 and Table 21. We can conclude that the biggest value is between the distributions for 306K and 303K.

In a further analysis, we compare distributions from signals of the RMSD obtained in the same way as a mean root square of end-to-end signals. Results for the Kullback–Leibler distance are presented in Table 22 and Table 23. We can conclude that for the time window 0–30 ns, the most significant value of differences is for the distribution for 309K and the distribution for 300K. For the time window 70–100 ns, the most significant value is for 306K and 303K.

In all tables of the Kullback–Leibler divergence measures we can see 0 value. For all these cases we compare two raw data from the same time window in the same temperature. Results for 306K we present in Table 24. We can see that the distribution of Kullback–Leibler distances are widely dispersed. We can observe a similar dispersion in other temperatures.

Referring to Figure 11, we observe simultaneous small valued events for the 0–30 ns simulation time window. There is no such events for the 70–100 ns simulation time window, see Figure 12. Here, the copula with the “lower tail dependency”, such as the Clayton, can provide the proper model of the RMSD for the 0–30 ns simulation time window.

## 3. Discussion

Albumin plays an essential role in many biological processes, such as the lubrication of articular cartilage. Hence, knowledge of its dynamics in various physiological conditions can help us to understand better its role in reducing friction. From a general point of view, albumin behaves stable regardless of the temperature. The angle energy of the protein fluctuates similarly regardless of the temperature (see Figure 4 and Figure 5). However, its mean value rises monotonically with temperature. On the other hand, the size exponent μ fluctuates with temperature. We calculate, using the Kruscal–Wallis test, that these fluctuations generate small values of *H* parameter and, as a result, indicate a hypothesis about the equality of medians.

Further, referring to Figure 6 and Figure 7, we can see that the dihedral energy seems to obey (at least at some range of simulation times) the power law-like relation, although the scaling exponent is relatively small. In Figure 7, we can see this relation in the range of temperature from 306*K*–309*K*. This exponent does not change in time *t* too much. It suggests stable behavior. We consider the conformational entropy of the system. For its plot versus simulation time see Figure 9 and Figure 10. On each graph, we can observe a pattern which seems to attain an oscillatory behavior. Fisher’s test allows describing this property by providing the *p*-value parameter. If this parameter, for some time series of entropy, is relatively small in contrast to *p*-value parameters for another time series of entropy, then the statistical hypothesis $H_{0}$:β=0 is less probable than in another case. In Table 14, we can observe that in the range of temperature 303*K*–309*K*, there is a local minimum of *p*-values. A similar effect can be observed in Table 15 when we calculate the mean *p*-value for each column. We can also observe that there is a local minimum of around 306K. A smaller mean *p*-value is seen for 300K and 312K. This means that both temperatures exhibit the analogical tendency to favor the hypothesis β≠0. We can also see this in Figure 9, where for temperatures 300K, 306K, and 312K, the regularity of periodicity increases more than at other temperatures. Other tests, including the Kruscal–Wallis test followed by the Conover–Iman multi-comparison test, Kullback–Leibler distance, and Kullback–Leibler divergence give consistent results, while referring to the mean RMSD of the protein of interest.

This globular protein is soluble in water and can bind both cations and anions. By analyzing Figure 1 and Figure 8, it is clear that albumin’s secondary structure is mainly standard α-helix. Charged amino acids (AA) have a large share in albumin’s structure, especially Aspartic and Glutamic acid (25% of all AA) and hydrophobic Leucine and Isoleucine (16%). This composition enables it to preserve its conformation, due to intramolecular interactions as well as interactions with the solvent. Two main factors are at play here, namely hydrophobic contacts and hydrogen bonds (which also occur between protein and water). A large number of charged AA result in a considerable number of inter- and intramolecular hydrogen bonds. On the other hand, hydrophobic AA result in a more substantial impact when the hydrophobic effect stabilizes a protein’s core. As shown by Rezaei-Tavirani et al. [29], the increase of temperature in the physiological range of temperatures results in conformational changes in albumin, which cause more positively charged molecules to be exposed. This, in turn, results in a lower concentration of cations near a molecule. Albumin is known to largely contribute to the osmotic pressure of plasma, where the presence of ions largely influences this property. An increase of temperature results in a reduction of the osmotic pressure of blood, which in feverish conditions can lead to a higher concentration of urine. Because water is the main component of blood and it has a high heat capacity, the increase of urine concentration results in better removal of heat from the body [29]. The difference in dynamics between temperatures shown in the present study could indicate albumin’s binding affinity with other SF components and thus changes its role in biolubrication toward anti-inflamatory. However, due to the high complexity of the fluid, more research has to be performed.

These chemical properties result in an effect of elasticity vs. swelling competition for albumin chains immersed in water. If the elasticity of the chain fully dominates over albumin’s swelling-induced counterpart (interactions of polymer beads and water with network/bond creation), the exponent would have a value of 1/2. If, in turn, a reverse effect applied, the value of the exponent should approach the value of 1/4, [30] (note the two first Eqs only). As a consequence, and what has not been discovered by any other study, the exponent obtained from the simulation in the examined range of temperature looks as if it is near an arithmetic mean of the two exponents mentioned, namely: μ=1/2(2/4+1/4)=3/8, thus pointing to 0.375∼0.4. The difference of about 0.025 very likely comes from the fact that the elastic effect on albumin shows up to win over the swelling-assisted counterterm(s), cf. the Hamiltonian used for the simulations. The exponent μ=3/8 is for itself called the De Gennes exponent, and it is reminiscent of the De Gennes dimension-dependent gelation exponent 2/(d+2), see [30] (eq 15 therein; d=3-case). It suggests that albumin’s elastic effect, centering at about 0.4, is fairly well supported by the internal network and thus supports the creation of bonds, which has been revealed by the present study. Of course, the overall framework is well-substantiated in terms of the scaling concept [3,7]. Regular oscillations of entropy suggest that the system oscillates around the equilibrium state. Furthermore, the Flory–De Gennes exponent appears to be unchanged for both observation windows. Therefore, we expect the system to be near the equilibrium state. We expect the dynamic of the system, for longer simulation times, to be similar to the presented one. The AMBER force field has been reported on overstabilizing helices in proteins. Therefore our next work will report on the effect of force field and water model used.

## 4. Materials and Methods

The structure of human albumin serum (code 1e78) has been downloaded from a protein data bank (https://www.rcsb.org/structure/1E78) as a starting point to simulations. We use the YASARA Structure Software (Vienna, Austria) [31] to perform MD simulations. Besides this, a three-site model (TIP3P) of water was used [32]. All-atom simulations were performed under the following conditions: temperature 310K, pH=7.0 and in 0.9% NaCl aqueous solution, with a time step of 2fs. Berendsen barostat and thermostat with a relaxation time of 1fs were used to maintain constant temperature and pressure. To minimize the inuence of rescaling, YASARA does not use the strongly fluctuating instantaneous microscopic temperature to rescale velocities at each simulation step (classic Berendsen thermostat). Instead, the scaling factor is calculated according to Berendsen’s formula from the time average temperature. The charge was –15. Simulations were carried for 100 ns, due to the fact that throughout this time very small fluctuations of entropy measured could be seen. All-atom molecular dynamics simulations were performed using the AMBER03 force field [33]. The AMBER03 potential function describing interactions among particles takes into account electrostatic, van der Waals, bond, bond angle, and dihedral terms:(17)$$\begin{array}{c} E_{total}=\sum_{bonds}k_{b}{\left(r-r_{eq}\right)}^{2}+\sum_{angle}k_{\phi }{\left(\phi -\phi_{eq}\right)}^{2}\\ \;+\sum_{dihedrals}\frac{V_{n}}{2}\left[1+\cos\left(n\psi -\gamma \right)\right]+\sum_{i<j}\left[\frac{A_{ij}}{r_{ij}^{12}}-\frac{B_{ij}}{r_{ij}^{6}}+\frac{q_{i}q_{j}}{\epsilon r_{ij}}\right] \end{array}$$
where, $k_{b}$ and $k_{\phi }$ are the force constants for the bond and bond angles, respectively; *r* and ϕ are bond length and bond angle; $r_{eq}$ and $\phi_{eq}$ are the equilibrium bond length and bond angle; ψ is the dihedral angle and $V_{n}$ is the corresponding force constant; the phase angle γ takes values of either $0^{\circ }$ or $180^{\circ }$. The non-bonded part of the potential is represented by van der Waals ($A_{ij}$) and London dispersion terms ($B_{ij}$) and interactions between partial atomic charges ($q_{i}$ and $q_{j}$). ϵ is the dielectric constant.

For each spherical angle distribution, we performed a normalized histogram consisting of 50 bins. Such a bin number is a compromise between a resolution and smoothness of histograms. Taking the empirical probability $p_{i}$ of data being in *i*-th bin, we estimate the entropy [18,34]:(18)$$S=-R_{0}\sum_{i}p_{i}\log\left(p_{i}\right),$$
where we use the gas constant $R_{0}=8.314\frac{J}{K\cdot mol}$. The sum goes over the discretized Ramachandran space. Obviously, since entropy is an information measure, it depends on a bin size, hence we have here only an estimation applicable for entropy comparisons for different temperatures and simulation times, since bin sizes are constant for all simulations and temperatures. Numerical data analyzes were performed in the Scilab development environment [35].

## 5. Conclusions

We present several scenarios of an analysis of simulations of the albumin dynamics. The albumin is the linear protein, which makes simulations and analysis straightforward. In particular, we get the end-to-end vector values, and we present that changes for exponent do not vary significantly. The more detailed analysis can be performed, however, using the entropy that appears to oscillate in the function of the simulation time. In general, these oscillations are regular, which is approved by employing the statistical test.

However, our findings show that near the 306K temperature, we can observe a local minimum of the *p*-value of this test. Interestingly, this minimum changes its position near the 312K temperature for the 70–100 ns time window. We also consider the RMSD parameter, which for the second time window, exhibits significant differences between the median of the sets defined for temperatures—we observe clustering.

When summarizing, let us note that the elastic energy terms included in the Hamiltonian (Equation (Equation 17)) provide a particularly robust bond vs. angle contribution to the elastic energy of the biopolymer, especially when arguing its role in the angles’ domain. The remaining non-elastic terms in Equation (Equation 17), in turn, contribute, in general, to the swelling assisted co-effect. Thus, the already mentioned Van der Waals and electrostatic terms are responsible for the overall attraction–repulsion behavior of the network-like swollen albumin, whereas the last “liquid–crystalline” type of energetic inclusion to Equation (Equation 13) samples the dihedral angles space, an observation already discussed above.

## Figures and Tables

**Figure 1 entropy-22-00405-f001:**
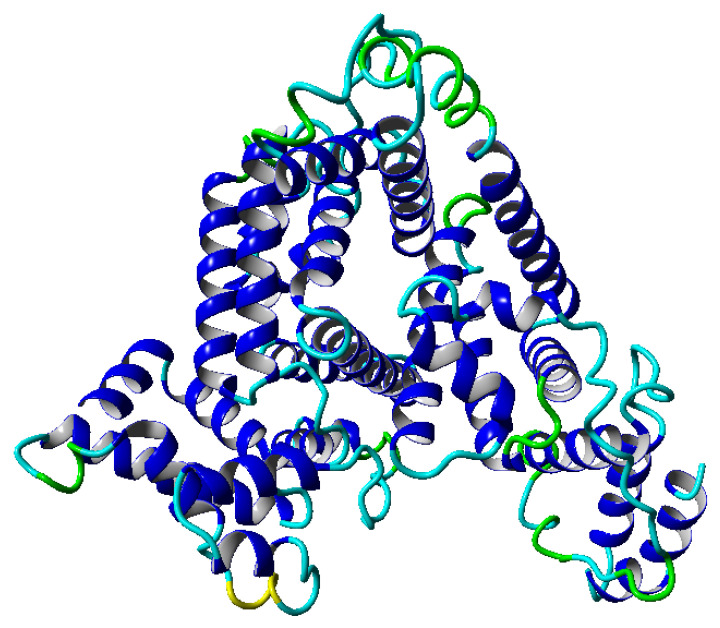
Structure of the albumin protein in a ribbon-like form. The colors on the molecular surface indicate the secondary structure: blue—α-helix, green—turn, yellow—3–10 helix, cyan—coil.

**Figure 2 entropy-22-00405-f002:**
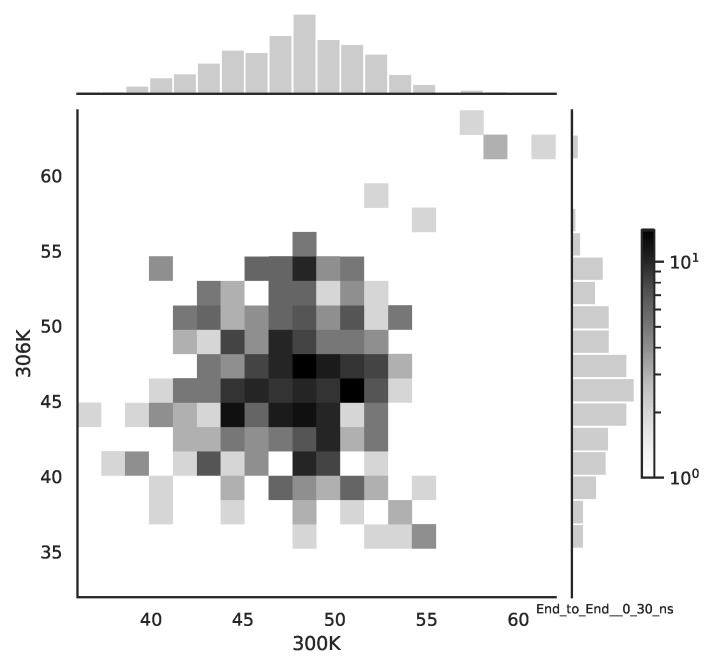
The bivariate histogram of chosen mean end-to-end signals in the 0–30 ns simulation time window.

**Figure 3 entropy-22-00405-f003:**
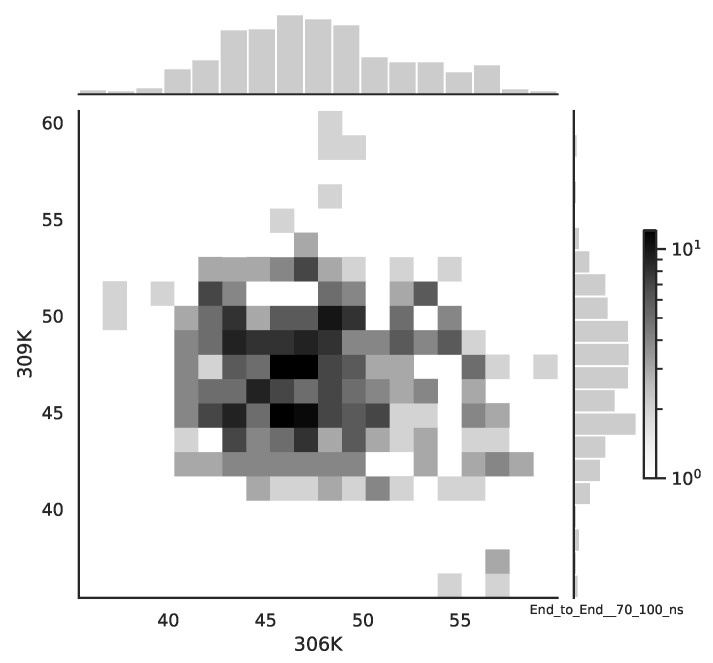
The bivariate histogram of chosen mean end-to-end signals in the 70–100 ns simulation time window.

**Figure 4 entropy-22-00405-f004:**
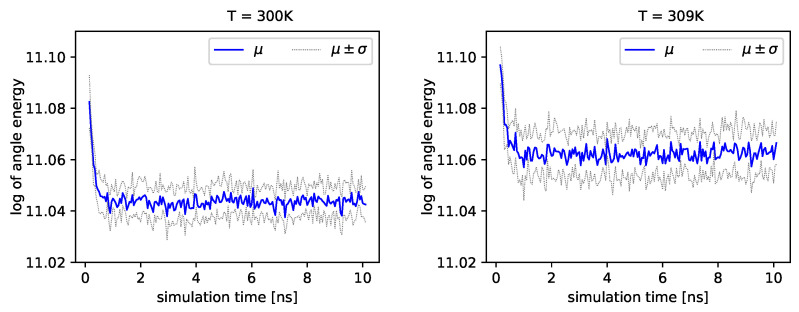
Logarithms of angle energies vs. the simulation time (0–10 ns), exemplary outcome of T=300K and T=309K.

**Figure 5 entropy-22-00405-f005:**
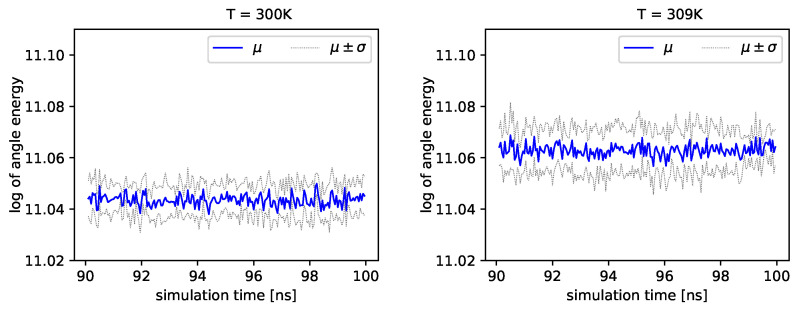
Logarithms of angle energies vs. the simulation time (90–100 ns), exemplary outcome of T=300K and T=309K.

**Figure 6 entropy-22-00405-f006:**
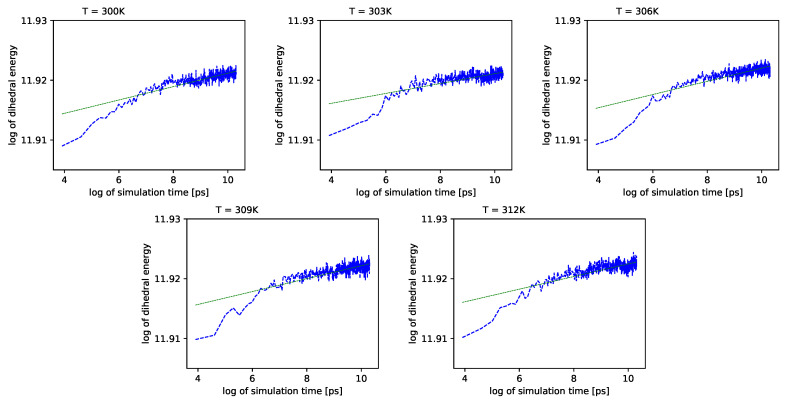
Dihedral energies for each temperature vs. simulation time. We use the double logarithmic scale. Green lines represent linear regression.

**Figure 7 entropy-22-00405-f007:**
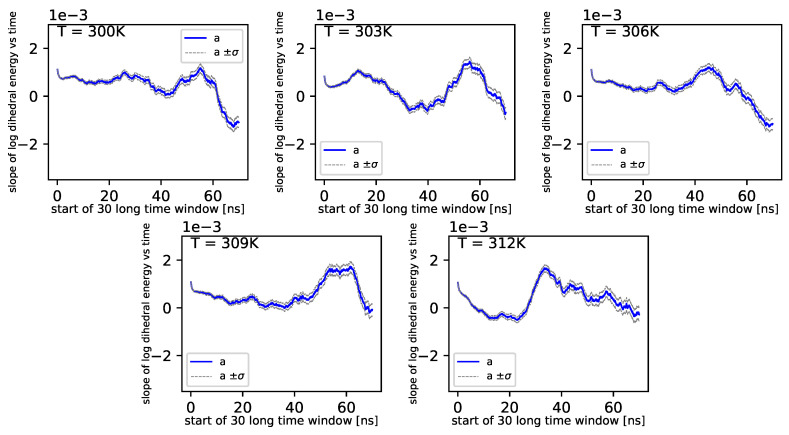
Slope of dihedral angle from regression.

**Figure 8 entropy-22-00405-f008:**
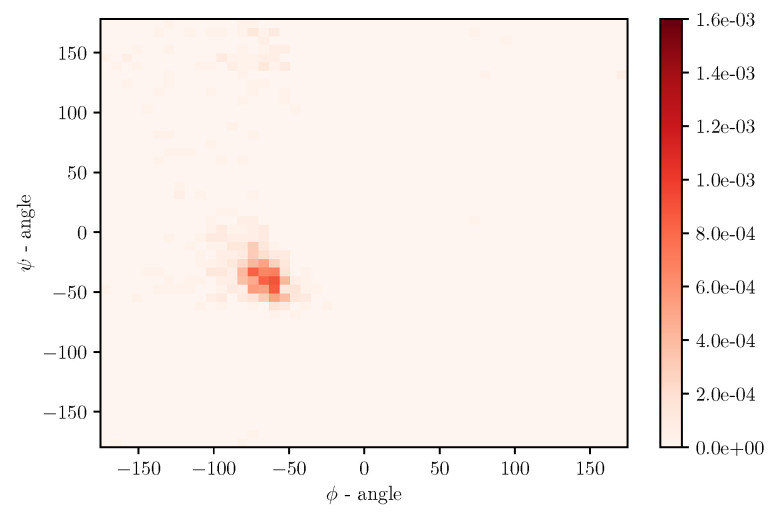
Numerically obtained distribution of angles ϕ and ψ for T=300K and 30 ns of simulation.

**Figure 9 entropy-22-00405-f009:**
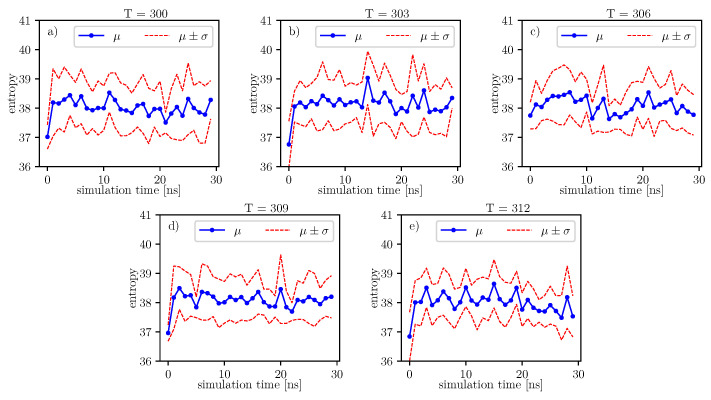
Value of conformational entropy for various temperatures and 0–30 ns of simulation time.

**Figure 10 entropy-22-00405-f010:**
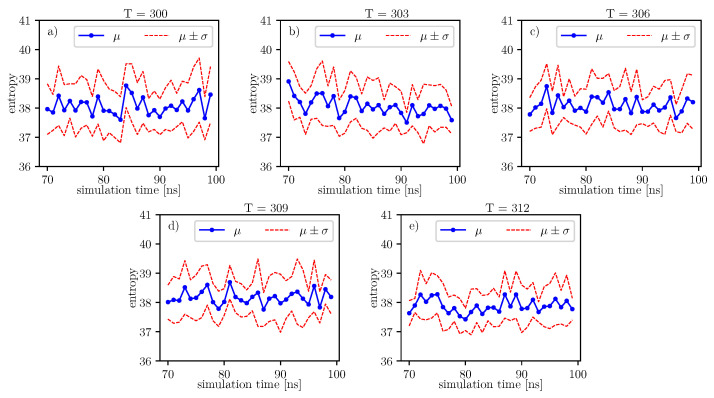
Value of conformational entropy for various temperatures and 70–100 ns of simulation time.

**Figure 11 entropy-22-00405-f011:**
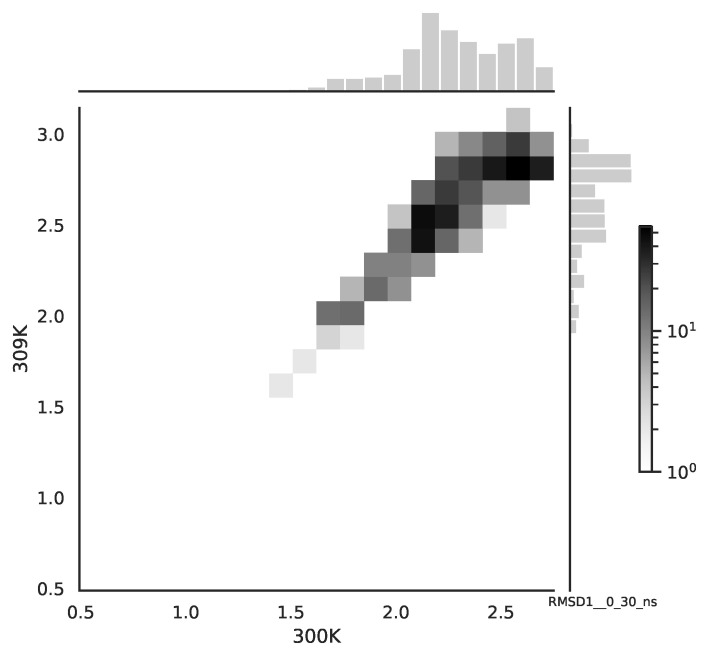
The bivariate histogram of the chosen mean RMSD signals in the 0–30 ns simulation time window.

**Figure 12 entropy-22-00405-f012:**
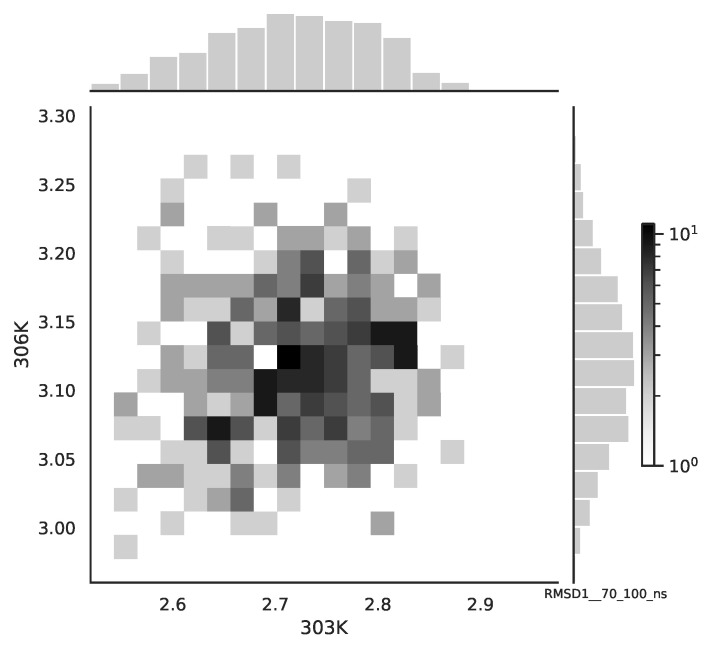
The bivariate histogram of the chosen mean RMSD signals in the 70–100 ns time window.

**Table 1 entropy-22-00405-t001:** Root mean square length of the end-to-end vector for 300K–312K.

Parameter	300K	303K	306K	309K	312K
${\langle R^{2}\rangle }^{1/2}$ [Å] for 0 ns–30 ns	47.88	46.69	46.72	47.09	47.81
${\langle R^{2}\rangle }^{1/2}$ [Å] for 70 ns–100 ns	47.12	46.67	47.72	47.13	47.64

**Table 2 entropy-22-00405-t002:** Values of *H* statistic in the Kruscal–Wallis test for two time windows for a root mean square end-to-end vector.

Window	Value of *H*
0–30 ns	H=0.62
70–100 ns	H=1.22

**Table 3 entropy-22-00405-t003:** Kullback–Leibler divergence for root mean square length of mean end-to-end signals in the time window 0–30 ns. The initial data is the same as for the Kruscal–Wallis test.

Temperature of the Reference Probability	300K	303K	306K	309K	312K
300K	0	0.9324	1.3858	0.8946	0.7604
303K	0.9324	0	1.4628	0.8176	1.6165
306K	0.2802	0.3703	0	0.4096	0.1532
309K	0.9062	0.8161	1.4816	0	2.4139
312K	0.2040	0.3710	0.7023	1.1887	0

**Table 4 entropy-22-00405-t004:** Symmetric Kullback–Leibler distance for a root mean square length of end-to-end signals in the time window 0–30 ns. The initial data is the same as for the Kruscal–Wallis test.

	300K	303K	306K	309K	312K
300K	0	0.9324	0.8330	0.9004	0.4822
303K	0.9324	0	0.9165	0.8169	0.9938
306K	0.8330	0.9165	0	0.9456	0.4278
309K	0.9004	0.8169	0.9456	0	1.8013
312K	0.4822	0.9938	0.4278	1.8013	0

**Table 5 entropy-22-00405-t005:** Kullback–Leibler divergence for a root mean square length of end-to-end signals in the time window 70–100 ns. The initial data is the same as for the Kruscal–Wallis test.

Temperature of the Reference Probability	300K	303K	306K	309K	312K
300K	0	0.1983	0.1983	0.1409	0.2883
303K	0.7474	0	0.8555	1.6675	2.6390
306K	0.7474	0.8555	0	1.6675	1.1576
309K	0.1351	0.4423	0.4423	0	0.1925
312K	0.8113	1.2084	1.8656	0.7532	0

**Table 6 entropy-22-00405-t006:** Symmetric Kullback–Leibler distance for a root mean square length of end-to-end signals in the time window 70–100 ns. The data is the same for the Kruscal–Wallis test.

	300K	303K	306K	309K	312K
300K	0	0.4728	0.4728	0.1380	0.5498
303K	0.4728	0	0.8555	1.0549	1.9237
306K	0.4728	0.8555	0	1.0549	1.5116
309K	0.1380	1.0549	1.0549	0	0.4728
312K	0.5498	1.9237	1.5116	0.4728	0

**Table 7 entropy-22-00405-t007:** Kullback–Leibler divergence between distributions of mean end-to-end signals in the time window 0–30 ns.

Temperature of the Reference Probability	300K	303K	306K	309K	312K
300K	0	0.1056	0.3194	0.1367	0.1689
303K	0.0939	0	0.2370	0.1261	0.1425
306K	0.2062	0.1450	0	0.1528	0.1841
309K	0.1276	0.1144	0.1877	0	0.2319
312K	0.1288	0.1275	0.2126	0.1848	0

**Table 8 entropy-22-00405-t008:** Kullback–Leibler distance for mean end-to-end signals in the time window 0–30 ns.

	300K	303K	306K	309K	312K
300K	0	0.0997	0.2628	0.1321	0.1489
303K	0.0997	0	0.1935	0.1202	0.1350
306K	0.2628	0.1935	0	0.1703	0.1984
309K	0.1321	0.1202	0.1703	0	0.2084
312K	0.1489	0.1350	0.1984	0.2084	0

**Table 9 entropy-22-00405-t009:** Kullback–Leibler divergence for mean end-to-end signals in the time window 70–100 ns.

Temperature of the Reference Probability	300K	303K	306K	309K	312K
300K	0	0.0920	0.1605	0.1824	0.1183
303K	0.1155	0	0.2969	0.1642	0.1474
306K	0.1108	0.1324	0	0.1621	0.1206
309K	0.1628	0.1266	0.2719	0	0.1206
312K	0.0886	0.0979	0.1356	0.1122	0

**Table 10 entropy-22-00405-t010:** Kullback–Leibler distance for mean end-to-end signals in the time window 70–100 ns.

	300K	303K	306K	309K	312K
300K	0	0.1038	0.1356	0.1726	0.1034
303K	0.1038	0	0.2147	0.1454	0.1226
306K	0.1356	0.2147	0	0.2170	0.1281
309K	0.1726	0.1454	0.2170	0	0.1164
312K	0.1034	0.1226	0.1281	0.1164	0

**Table 11 entropy-22-00405-t011:** Kullback–Leibler distance for raw data end-to-end signals in the time window 70–100 ns and 306K.

Number of Molecule	1	2	3	4	5	6	7	8	9
1	0	0.2927	0.2073	0.2156	0.2450	0.3016	0.1751	0.2072	0.4298
2	0.2927	0	0.1661	0.0919	0.0931	0.1589	0.2367	0.1310	0.2765
3	0.2073	0.1661	0	0.1347	0.2135	0.3059	0.1282	0.1013	0.1843
4	0.2156	0.0919	0.1347	0	0.1203	0.1353	0.1499	0.1338	0.2585
5	0.2450	0.0931	0.2135	0.1203	0	0.1478	0.2635	0.1434	0.3083
6	0.3016	0.1589	0.3059	0.1353	0.1478	0	0.2812	0.1978	0.3122
7	0.1751	0.2366	0.1282	0.1499	0.2635	0.2812	0	0.1318	0.2281
8	0.2072	0.1310	0.1013	0.1338	0.1434	0.1978	0.1318	0	0.1614
9	0.4298	0.2765	0.1843	0.2585	0.3083	0.3122	0.2281	0.1615	0

**Table 12 entropy-22-00405-t012:** Size exponent obtained according to Formula (Equation 5).

Parameter	300K	303K	306K	309K	312K
Size exponent (μ) for 0 ns–30 ns	0.3986	0.3947	0.3948	0.3960	0.3984
Size exponent (μ) for 70 ns–100 ns	0.3961	0.3946	0.3981	0.3961	0.3978

**Table 13 entropy-22-00405-t013:** Values of *H* statistic in the Kruscal–Wallis test for μ parameter for two time windows.

Window	Value of Parameter *H*
0–30 ns	H=0.48
70–100 ns	H=1.47

**Table 14 entropy-22-00405-t014:** Probability (*p*-value) for mean entropy over time in the time interval 0–30 ns.

	300K	303K	306K	309K	312K
*p*-value	0.849	0.601	0.009	0.536	0.143

**Table 15 entropy-22-00405-t015:** Probability (*p*-value) for entropy of single trajectory over time in the time interval 0–30 ns.

Number of Signal	300K	303K	306K	309K	312K
1	0.439	0.737	0.367	0.754	0.643
2	0.153	0.660	0.961	0.922	0.835
3	0.320	0.483	0.012	0.558	0.0551
4	0.263	0.372	0.700	0.223	0.323
5	0.544	0.657	0.057	0.130	0.066
6	0.114	0.385	0.232	0.416	0.360
7	0.499	0.529	0.008	0.006	0.007
8	0.013	0.871	0.610	0.692	0.456
9	0.245	0.891	0.871	0.832	0.694
mean	0.288	0.621	0.424	0.504	0.382

**Table 16 entropy-22-00405-t016:** Probability (*p*-value) for entropy of a single trajectory over time in the time interval 70–100 ns.

Number of Signal	300K	303K	306K	309K	312K
1	0.328	0.720	0.460	0.078	0.041
2	0.183	0.890	0.146	0.713	0.898
3	0.092	0.378	0.219	0.464	0.264
4	0.764	0.800	0.440	0.629	0.297
5	0.030	0.802	0.275	0.323	0.313
6	0.220	0.108	0.330	0.440	0.540
7	0.279	0.121	0.382	0.253	0.046
8	0.582	0.299	0.976	0.324	0.457
9	0.438	0.550	0.438	0.951	0.642
mean	0.324	0.519	0.407	0.464	0.389

**Table 17 entropy-22-00405-t017:** Results of the Conover–Iman multi-comparison test in the time interval 70–100 ns.

Set of Molecule	300K	303K	306K	309K	312K
300K	-	1.2222	16.1111	12.5556	10.3333
303K	1.2222	-	17.3333	13.7778	11.5556
306K	16.1111	17.3333	-	3.5556	5.7778
309K	12.5556	13.7778	3.5556	-	2.2222
312K	10.3333	11.5556	5.7778	2.2222	-

**Table 18 entropy-22-00405-t018:** Kullback–Leibler divergence for the mean RMSD in the time window 0–30 ns. The data are the same as for the Kruscal–Wallis test.

Temperature of Reference Distribution	300K	303K	306K	309K	312K
300K	0	0.4045	2.0767	2.917	0.1932
303K	1.6280	0	1.6853	2.4913	0.6891
306K	3.3967	2.6057	0	1.8597	2.3939
309K	2.2053	0.5003	1.1502	0	1.1444
312K	0.1598	0.1343	1.6845	2.5806	0

**Table 19 entropy-22-00405-t019:** Symmetric Kullback–Leibler distance for the mean RMSD in the time window 0–30 ns. The data are the same as for the Kruscal–Wallis test.

	300K	303K	306K	309K	312K
300K	0	1.0162	2.7367	2.5600	0.1765
303K	1.1016	0	2.1455	1.4958	0.4117
306K	2.7367	2.1455	0	1.5049	2.0392
309K	2.5600	1.4958	1.5049	0	1.8625
312K	0.1765	0.4117	2.0392	1.8625	0

**Table 20 entropy-22-00405-t020:** Kullback–Leibler divergence for a mean RMSD in the time window 70–100 ns. The data are the same as for the Kruscal–Wallis test.

Temperature of Reference Distribution	300K	303K	306K	309K	312K
300K	0	0.2941	2.7930	2.1527	3.9428
303K	0.8374	0	5.1756	2.2304	3.7127
306K	1.3305	2.4804	0	0.8902	2.8295
309K	4.0173	3.3975	3.3421	0	2.7354
312K	3.6680	2.2372	3.3862	0.8271	0

**Table 21 entropy-22-00405-t021:** Symmetric Kullback–Leibler distance for the mean RMSD in the time window 70–100 ns. The data are the same as for the Kruscal–Wallis test.

	300K	303K	306K	309K	312K
300K	0	0.5657	2.0617	3.0850	3.8054
303K	0.5657	0	3.8280	2.8139	2.9749
306K	2.0617	3.8280	0	2.1161	3.1078
309K	3.0850	2.8139	2.1161	0	1.7812
312K	3.8054	2.9749	3.1078	1.7812	0

**Table 22 entropy-22-00405-t022:** Symmetric Kullback–Leibler distance for signals of the RMSD in the time window 0–30 ns.

	300K	303K	306K	309K	312K
300K	0	0.3686	0.7090	1.7109	0.1870
303K	0.3686	0	0.3051	1.2909	0.1838
306K	0.7090	0.3051	0	0.7468	0.5706
309K	1.7109	1.2909	0.7468	0	1.4513
312K	0.1870	0.1838	0.5706	1.4513	0

**Table 23 entropy-22-00405-t023:** Symmetric Kullback–Leibler distance for signals of the RMSD in the time window 70–100 ns.

	300K	303K	306K	309K	312K
300K	0	0.1056	6.3487	5.5232	3.9800
303K	0.1056	0	6.4065	5.7737	4.3824
306K	6.3487	6.4065	0	0.9648	1.7829
309K	5.5232	5.7737	0.9648	0	0.4260
312K	3.9800	4.3824	1.7829	0.4260	0

**Table 24 entropy-22-00405-t024:** Symmetric Kullback–Leibler distance for raw signals of the RMSD in the time window 70–100 ns for 306K.

Number of Molecule	1	2	3	4	5	6	7	8	9
1	0	1.669	0.388	0.277	0.137	0.500	3.260	0.560	0.249
2	1.669	0	1.974	2.565	2.074	1.615	0.660	1.498	1.410
3	0.388	1.974	0	0.241	0.617	0.246	3.589	0.299	0.352
4	0.277	2.565	0.241	0	0.250	0.494	3.867	0.628	0.406
5	0.137	2.074	0.617	0.250	0	0.540	3.593	0.597	0.275
6	0.500	1.615	0.246	0.494	0.540	0	3.338	0.080	0.140
7	3.257	0.660	3.589	3.867	3.593	3.337	0	3.118	2.795
8	0.560	1.498	0.300	0.628	0.597	0.080	3.118	0	0.165
9	0.249	1.410	0.352	0.406	0.275	0.140	2.795	0.165	0

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
