# Peer review of "Changes of Conformation in Albumin with Temperature by Molecular Dynamics Simulations"

_entropy, 2020, doi:10.3390/e22040405_

Round 1
Reviewer 1 Report
Weber et al. conducted a simple theoretical study of the conformational preferences of the protein albumin at different temperatures. Using all-atom molecular dynamics simulations, the team analysed the global dynamic properties of the protein through angle calculations and conformational entropy. Some non-standard validation/analysis techniques were also employed such as Fisher’s test, Kruscal-Wallis test and Flory exponent analysis, to identify the small differences in results due to temperature fluctuations.
While the study is logical and well presented, there several concerns that need to be addressed before consideration for publication:
- Albumin is one of the more widely studied proteins, theoretically and experimentally. I found the manuscript very weak in validating or comparing their findings with the available literature on albumin. It is not clear what behaviour is the protein expected to exhibit at these temperatures?
- The forcefield selected for this study is Amber03, have the authors explored the role of the forcefield here? Amber is known to overstabilise helices so it is no surprise that the authors did not see any appreciable differences between the different temperatures. Once again, comparison and validation with ideally experimental literature is needed
- Rather than looking at the actual plots, it’s important to state the length of the simulations in the Materials and Methods section as well. 100ns of simulations is insufficiently long for such a large protein. How did the authors ensure they have sampled wider conformational space? They could consider running several simulations starting from different initial velocities or starting orientations?
- What is the total charge of albumin in the simulations? Were any counterions used? Salts? If this is to be considered a study conducted in physiological solution, salts must be included. How were the ends of the protein also terminated?
- A careful proofread would also be beneficial; eg. pg. 3 “hipotesis”
Author Response
Dear Reviewer,
We have carefully considered your comments and detail our response below. We hope to see our manuscript published in the special issue of Entropy, and thank you for your remarks as they improved the manuscript significantly.
Piotr Bełdowski
On behalf of all co-authors
Weber et al. conducted a simple theoretical study of the conformational preferences of the protein albumin at different temperatures. Using all-atom molecular dynamics simulations, the team analysed the global dynamic properties of the protein through angle calculations and conformational entropy. Some non-standard validation/analysis techniques were also employed such as Fisher’s test, Kruscal-Wallis test and Flory exponent analysis, to identify the small differences in results due to temperature fluctuations.
While the study is logical and well presented, there several concerns that need to be addressed before consideration for publication:
Comment:
Albumin is one of the more widely studied proteins, theoretically and experimentally. I found the manuscript very weak in validating or comparing their findings with the available literature on albumin. It is not clear what behaviour is the protein expected to exhibit at these temperatures?
Response:
The power-law exponents for oligomers span a narrow range of 0.38–0.41
https://www.ncbi.nlm.nih.gov/pmc/articles/PMC5053138/#!po=32.3529
which fits results obtained with our simulations. This reference was added to the manuscript.
There is also another experimental study showing albumin’s behaviour in the temperature range close to ours. We have added 3 new Literature positions referring to the obtained results in terms of current state of knowledge.
Comment:
The forcefield selected for this study is Amber03, have the authors explored the role of the forcefield here? Amber is known to overstabilise helices so it is no surprise that the authors did not see any appreciable differences between the different temperatures. Once again, comparison and validation with ideally experimental literature is needed
Response:
The effect of force field and water model used are two factors that will be studied in the follow up studies as they play a major role on how realistic the obtained results are. We commented on this in Discussion section.
Comment:
Rather than looking at the actual plots, it’s important to state the length of the simulations in the Materials and Methods section as well. 100ns of simulations is insufficiently long for such a large protein. How did the authors ensure they have sampled wider conformational space? They could consider running several simulations starting from different initial velocities or starting orientations?
Response:
We were For each temperature we performed 10 simulations with different initial velocities which were different for each simulation in particular temperature. Simulations were carried for 100 ns due to the fact that throughout this time very small fluctuations of entropy measured could be seen.
Comment:
What is the total charge of albumin in the simulations? Were any counterions used? Salts? If this is to be considered a study conducted in physiological solution, salts must be included. How were the ends of the protein also terminated?
Response:
Protein charge was -15 as the pH of the solvent was 7.0. The simulations were performed for 0.9% NaCl water solution. The structure was taken from the protein data bank and the code for the molecule used is mentioned in the text.
Comment:
A careful proofread would also be beneficial; eg. pg. 3 “hipotesis”
Response:
Thank you, we have carefully checked the text this time. Native speaker have check the text to fix the problem.
Reviewer 2 Report
The authors present a detailed analysis of the conformational behaviour of a model protein, albumin, depending on the temperature. A range of temperatures in the "physiological range (300-312 K) is employed. The analysis of the variations in the key torsion angles of the backbone is performed and the conformational entropy is estimated. The manuscript is technically sound. However, regarding the possible applications (drug design, as mentioned by the authors), the behaviour of the side chains should also be explored. Obviously, this is a major task, but it should be important to dissect their behaviour with temperature. Moreover, there is one more detail. The use of the TIP3P model for water has some limitations and some properties are better modelled by the TIP5P alternative. A comment on this should also be welcomed.
Author Response
Dear Reviewer,
We have carefully considered your comments and detail our response below. We hope to see our manuscript published in the special issue of Entropy, and thank you for your consideration.
Piotr Bełdowski
On behalf of all co-authors
The authors present a detailed analysis of the conformational behaviour of a model protein, albumin, depending on the temperature. A range of temperatures in the "physiological range (300-312 K) is employed. The analysis of the variations in the key torsion angles of the backbone is performed and the conformational entropy is estimated. The manuscript is technically sound. However, regarding the possible applications (drug design, as mentioned by the authors), the behaviour of the side chains should also be explored. Obviously, this is a major task, but it should be important to dissect their behaviour with temperature.
Comment:
Moreover, there is one more detail. The use of the TIP3P model for water has some limitations and some properties are better modelled by the TIP5P alternative. A comment on this should also be welcomed.
Response:
We used the TIP3P model as it is widely used in biophysical molecular dynamics simulations due to its quite good replication of water properties and computationally very efficient when compared to e.g. TIP5P model. TIP5P is obviously better model and we plan to use it in future. However, YASARA offers only TIP3P model to be used. In our future work we plan to switch to different water model for more realistic simulations.
Reviewer 3 Report
The authors study the dynamics of albumen over a temperature range of 300K to 312K. The goal of the study is to see how the dynamics of this protein may change due to changes in T under physiological condistions, such as friction induced heating in synovial fluid. This an interesting idea, and the authors have performed some worthwhile simulations using the program YASARA.
However the paper is severely weakened by the choice of analysis techniques. Molecule dynamics provide atomic resolution information. Specifically one can get fine resolution well populations probability distributions for almost any structural or dynamic property, obtain time correlation functions etc. Yet the authors choose to use very weak, problematic tests from ‘classical statistics’ such as significance testing, non-parametric Kruschal-Wallis metrics. Many of their results are presented as significant/nonsignificant.
The problems with significance testing have been exhaustively aired in the stats literature (e.g. Greenland, Sander, Stephen J. Senn, Kenneth J. Rothman, John B. Carlin, Charles Poole, Steven N. Goodman, and Douglas G. Altman. “Statistical Tests, P Values, Confidence Intervals, and Power: A Guide to Misinterpretations.” European Journal of Epidemiology 31, no. 4 (2016): 337–50. https://doi.org/10.1007/s10654-016-0149-3.
Wasserstein, Ronald L., Allen L. Schirm, and Nicole A. Lazar. “Moving to a World Beyond ‘p < 0.05.’” The American Statistician 73, no. sup1 (March 29, 2019): 1–19. https://doi.org/10.1080/00031305.2019.1583913.).
The problem here is that with enough sampling (which MD can always provide) any small difference can be made significant. What matters is the size of the difference, and credible intervals. Take the end-to-end distribution, or radius of gyration obtained from an MD simulation. One could compare the distribution of these metrics using for example more powerful and informative information-theoretic measures such as Kulback-Liebler divergence. There is a good literature on analysis of MD simulations of proteins which the authors should consult. Their whole analysis section needs to be greatly augmented and modernized with current MD-analysis standard techniques.
Author Response
Dear Reviewer,
We have carefully considered your comments and detail our response below. We hope to see our manuscript published in the special issue of Entropy, and thank you for your remarks as they improved the manuscript significantly.
Piotr Bełdowski
On behalf of all co-authors
The authors study the dynamics of albumen over a temperature range of 300K to 312K. The goal of the study is to see how the dynamics of this protein may change due to changes in T under physiological condistions, such as friction induced heating in synovial fluid. This an interesting idea, and the authors have performed some worthwhile simulations using the program YASARA.
Comment:
However the paper is severely weakened by the choice of analysis techniques. Molecule dynamics provide atomic resolution information. Specifically one can get fine resolution well populations probability distributions for almost any structural or dynamic property, obtain time correlation functions etc. Yet the authors choose to use very weak, problematic tests from ‘classical statistics’ such as significance testing, non-parametric Kruschal-Wallis metrics. Many of their results are presented as significant/nonsignificant.
The problems with significance testing have been exhaustively aired in the stats literature (e.g. Greenland, Sander, Stephen J. Senn, Kenneth J. Rothman, John B. Carlin, Charles Poole, Steven N. Goodman, and Douglas G. Altman. “Statistical Tests, P Values, Confidence Intervals, and Power: A Guide to Misinterpretations.” European Journal of Epidemiology 31, no. 4 (2016): 337–50. https://doi.org/10.1007/s10654-016-0149-3.
Wasserstein, Ronald L., Allen L. Schirm, and Nicole A. Lazar. “Moving to a World Beyond ‘p < 0.05.’” The American Statistician 73, no. sup1 (March 29, 2019): 1–19. https://doi.org/10.1080/00031305.2019.1583913.).
The problem here is that with enough sampling (which MD can always provide) any small difference can be made significant. What matters is the size of the difference, and credible intervals. Take the end-to-end distribution, or radius of gyration obtained from an MD simulation. One could compare the distribution of these metrics using for example more powerful and informative information-theoretic measures such as Kulback-Liebler divergence. There is a good literature on analysis of MD simulations of proteins which the authors should consult. Their whole analysis section needs to be greatly augmented and modernized with current MD-analysis standard techniques
Response:
Thank you. We have added Kulback-Liebler divergence calculations in the text and commented on them. In the text you can find new Tables 3-10 with a newly obtained results for Kullback-Leibler tests. “We can also see this in Figure 7, where for temperatures 300K, 306K and 312K, the regularity of periodicity increases more than at other temperatures. Other tests, including the Kruscal-Wallis test followed by the Conover- Iman multi-comparison test, Kullback-Leibler distance, and Kullback-Leibler divergence give consistent results”.
Round 2
Reviewer 3 Report
The analysis of the authors MD simulations is much improved with the addition of the K-L divergence measure. However, two additional aspects should be addressed by the authors:
1) In the table of K-L divergence measures, the diagonal terms are by construction 0 since the authors would simply be comparing a distribution with itself. A more meaningful comparison is to compare two completely independent (i.e. starting with different randomized velocities, and independent equilibration/production) simulations e.g. 303K (simulation 1) with 303K (simulation 2) etc. (note NOT two different time slices of the same simulation) The K-L divergences would not be identically zero, but would provide a measure of 'reproducibility' for what the authors are looking at, and a scale for the divergences that they do see between 303K and 30K etc.
2) Measures of difference (statistical or information theory) are not themselves an end, but signposts for further analysis and physical interpretation. For the systems showing the largest difference, where is this coming from? is it from tails of the distribution (i.e. rare events, and therefore of further interest)) or simply differences spread across the distribution (probably less interesting). Are the differences localized in dihedral angles located certain parts of the structure (again potentially interesting) or spread across the structure. Figs. 2-5 don't really show much that can be interpreted in terms of structure/dynamics of albumin as a functinally interesting protein. The answers to these Q's lie in the simulations the authors have already performed, if they would dig a little deeper in their analysis. This would make a paper that basically says "Look, dynamics is different" into one that says something insightful about albumin
Author Response
Dear Reviewer,
The authors appreciate the constructive comments and the good suggestions of the reviewers. Our manuscript, "Changes of conformation in albumin with temperature by molecular dynamics simulations", has been revised following the reviewers' ideas.
The reviewer's comment: 1.
In the table of K-L divergence measures, the diagonal terms are by construction 0 since the authors would simply be comparing a distribution with itself. A more meaningful comparison is to compare two completely independent (i.e. starting with different randomized velocities, and independent equilibration/production) simulations e.g. 303K (simulation 1) with 303K (simulation 2) etc. (note NOT two different time slices of the same simulation) The K-L divergences would not be identically zero, but would provide a measure of 'reproducibility' for what the authors are looking at, and a scale for the divergences that they do see between 303K and 30K etc.
The authors' reply:
We obtain KL measures between every two distributions following from independent simulations of the molecule (molecules numbered from 1 to 9) for the same time window and the same temperature. According to our previous analyses, we tested distributions of raw data: End-to-end parameter and RMSD parameter. We inserted into the manuscript only two selected analyses in time window 70-100ns and temperature 306K. The rest of the results look similar. We can see that distances have random values, so we can assume that randomization of the initial values is fulfilled.
According to the reviewer's comment about randomized velocities, we checked input data to Fisher's test. Therefore we now have nine molecules in the Fisher test, not ten. In previous analyses, the 10-th molecule had initial velocities not taken in a random way, so it was the reason not to take it into account. We can see that for mean entropy beta, not equal zero in temperature 306K is now more evident than in previous analysis and another temperature.
The reviewer's comment:2.
Measures of difference (statistical or information theory) are not themselves an end, but signposts for further analysis and physical interpretation. For the systems showing the largest difference, where is this coming from? is it from tails of the distribution (i.e. rare events, and therefore of further interest)) or simply differences spread across the distribution (probably less interesting). Are the differences localized in dihedral angles located certain parts of the structure (again potentially interesting) or spread across the structure. Figs. 2-5 don't really show much that can be interpreted in terms of structure/dynamics of albumin as a functionally interesting protein.
The answers to these Q's lie in the simulations the authors have already performed, if they would dig a little deeper in their analysis. This would make a paper that basically says "Look, dynamics is different" into one that says something insightful about albumin
The authors' reply:
We analyze the bivariate histograms of mean RMSD signals in the 0-30ns and 70-100ns time window. We also analyze the bivariate histograms mean end-to-end signals in the 0-30ns and 70-100ns time windows.
We have shown that our results are consistent with research Rezaei-Tavirani et al. [27]. He shows that the increase of temperature in the physiological range of temperatures results in conformational changes in albumin, which cause more positively charged molecules to be exposed. This, in turn, results in a lower concentration of cations near a molecule. Here we supplement these studies by given global characteristics of these dynamics of conformation.